# Assessing the feasibility of an integrated collection of education modules for fall and fracture prevention (iCARE) for healthcare providers in long term care: A longitudinal study

Isabel B. Rodrigues[1,2], George Ioannidis[1], Lauren L. Kane[1], Loretta M. Hillier[2], Jonathan Adachi[1], George Heckman[3,4,5], John Hirdes[5], Jayna Holroyd-Leduc[6], Susan Jaglal[7], Sharon Kaasalainen[8], Sharon Marr[1], Caitlin McArthur[9], Sharon Straus[10], Jean-Eric Tarride[8,11,12], Momina Abbas[13], Andrew P. Costa[5,13], Arthur N. Lau[1], Lehana Thabane[13,14,15], Alexandra Papaioannou[1,13]*

1 Department of Medicine, McMaster University, Hamilton, Ontario, Canada, 2 Department of Community Health, University of Manitoba, Winnipeg, Manitoba, Canada, 3 Department of Medicine, Western University, London, Ontario, Canada, 4 Lawson Health Research Institute, St. Joseph's Health Care London, London, Ontario, Canada, 5 School of Public Health Sciences, University of Waterloo, Waterloo, Ontario, Canada, 6 Department of Medicine and Community Health Sciences, University of Calgary, Calgary, Alberta, Canada, 7 Department of Physical Therapy, University of Toronto, Toronto, Ontario, Canada, 8 School of Nursing, McMaster University, Hamilton, Ontario, Canada, 9 School of Physiotherapy, Dalhousie University, Halifax, Nova Scotia, Canada, 10 Department of Medicine, University of Toronto, Toronto, Ontario, Canada, 11 Center for Health Economics and Policy Analysis (CHEPA), McMaster University, Hamilton, Ontario, Canada, 12 The Research Institute of St. Joe's Hamilton, Programs for Assessment of Technology in Health (PATH), St. Joseph's Healthcare Hamilton, Hamilton, Ontario, Canada, 13 Department of Health Research Methods, Evidence and Impact, McMaster University, Hamilton, Ontario, Canada, 14 Biostatistics Unit, St. Joseph's Healthcare Hamilton, Hamilton, Ontario, Canada, 15 Faculty of Health Sciences, University of Johannesburg, Johannesburg, South Africa

* papaioannou@hhsc.ca

## Abstract

Falls and hip fractures are a major health concern among older adults in long term care (LTC) with almost 50% of residents experiencing a fall annually. Hip fractures are one of the most important and frequent fall-related injuries in LTC. There is moderate to strong certainty evidence that multifactorial interventions may reduce the risk of falls and fractures; however, there is little evidence to support its implementation. The purpose of this study was to determine the feasibility (recruitment rate and adaptations) with a subobjective to understand facilitators to and barriers of implementing the PREVENT (Person-centred Routine Fracture PreEVENTion) model in practice. The model includes a multifactorial intervention on diet, exercise, environmental adaptations, hip protectors, medications (including calcium and vitamin D), and medication reviews to treat residents at high risk of fracture. Our secondary outcomes were to determine if there was a change in knowledge uptake of the guidelines among healthcare providers and in the proportion of fracture prevention prescriptions post-intervention. We conducted a mixed-methods longitudinal cohort study in three LTC

**Data availability statement:** Data is available in the supplementary documents (please see S3 File and S5 Table).

**Funding:** The authors would like to thank the following funding agents for their support with the project. The lead author (IBR) was funded by the McMaster Institute for Research on Aging (MIRA), AGE-WELL, the Hamilton Health Sciences New Investigator Fund, the Canadian Institutes of Health Research Postdoctoral Award. The corresponding author (AP) received funding from the Amgen Competitive Grant Program in Bone Research Award. The funders played no part in developing the research design, collecting data collection, analyzing the results, or writing the manuscript.

**Competing interests:** This work was supported by the Amgen Competitive Grant Program in Bone Research Award. This does not alter our adherence to PLOS Global Public Health policies on sharing data and materials. There are no patents, products in development or marketed products associated with this research to declare.

homes across southern Ontario. A local champion was selected to help guide the implementation of the model and promote best practices. We reported recruitment rates using descriptive statistics and challenges to implementation using content analysis. We reported changes in knowledge uptake and in the proportion of fracture prevention medications using the McNemar's test. We recruited three LTC homes and identified one local champion for each home. We required two months to identify and train the local champion over three, 1.5-hour train-the-trainer sessions, and the local champion required three months to deliver the intervention to a team of healthcare professionals. We identified several facilitators, barriers, and adaptations to PREVENT. Benefits of the model include easy access to the Fracture Risk Scale (FRS), clear and succinct educational material catered to each healthcare professional, and an accredited Continuing Medical Educational module for physicians and nurses. Challenges included misperceptions between the differences in fall and fracture prevention strategies, fear of perceived side effects associated with fracture prevention medications, and time barriers with completing the audit report. Our study found an increase knowledge uptake of the guidelines and an increase in the proportion of fracture prevention prescriptions post-intervention.

## Introduction

Falls and hip fractures are a major health concern among older adults [1]. An estimated one in three individuals aged 65 years and older fall each year, and almost half of individuals living in long term care (LTC) fall annually [2]. Hip fractures in residents living in LTC are devastating and are likely to increase with current demographic trends in aging [3–5]. Each year, almost 50% of residents in LTC facilities fall at least once, and 40% of residents fall twice or more [4,6]. Of those LTC residents that fall, 10% to 25% of falls are associated with serious injuries requiring medical treatment with 2% to 6% resulting in fractures of the hip, wrist, or vertebra [7–9]. Hip fractures are particularly devastating as both mobility and the capacity to live independently are affected [4,7–9]. Moreover, about half of the residents in LTC who experience a hip fracture may die within the next year or develop total dependence within six months of their hip fracture [4,7–9]. Thus, interventions to reduce falls and fractures in LTC are important to decrease morbidity and mortality.

In 2015, the Canadian recommendations for preventing fractures in LTC were released to facilitate the evidence-based decision-making process [10]. Indeed, guidelines have the capacity to promote high quality practice informed by evidence, enable appropriate resource allocation, and advance research by identifying knowledge gaps [11]; however, the existence of guidelines alone is not enough to change practice [11,12]. The lack of guideline uptake in practice is evident with the high morbidity and mortality associated with fractures in LTC [13]. The major gap is the limited evidence on effective knowledge translation strategies to prevent falls and fragility fractures in residents who are at high risk of fractures [13–15]. Active strategies that utilize an integrated knowledge translation approach may help to uptake guidelines into practice [13]. Current methods to uptake guidelines involve co-developing models with end-users and stakeholders and piloting their implementation under real-world conditions.

PREVENT (Person-centred Routine Fracture PreEVENTion) is an educational outreach model for delivering routine assessments on fall and fracture prevention to healthcare providers working in LTC. The model includes a multifactorial educational intervention on diet,

exercise, environmental adaptations to reduce falls, hip protectors, medication (including calcium and vitamin D), and medication reviews. The educational material in the model aligns with the 2015 Canadian evidence-based recommendations [10] and with recommendations from Australia and the USA [16]. The model utilizes a multifactorial knowledge mobilization strategy that engages the entire multi-disciplinary LTC team (e.g., physicians, nurses, physiotherapists, dietitians, personal support workers) and builds automated identification and treatment of high-risk fractures into routine care processes.

The PREVENT model may have several benefits including reducing falls and fractures and improving quality of life of residents in LTC. Before a trial can be implemented to determine the effectiveness of the PREVENT model in practice, we need to determine the feasibility of implementing the model under real-world conditions. The purpose of the iCare study was to determine the feasibility of implementing the PREVENT model, herein known as PREVENT, to healthcare providers in LTC. The iCare study is the feasibility study and project name. Our secondary outcomes were to determine if PREVENT improved knowledge uptake of the Canadian recommendations [10] and increased the proportion of fracture prevention prescriptions post-intervention. We selected the proportion of change in fracture prevention medications as our secondary outcome since fracture prevention medications currently have the highest certainty of evidence to reduce the risk and the rate of fractures [17,18].

## Methods

### Study design

We conducted a longitudinal cohort study in three LTC homes across Ontario. We conducted our study in accordance with the STROBE 2007 guidelines (S1 Table) [19,20] and TIDieR 2014 checklist (S2 Table) [21]. We registered our study on clinicaltrials.gov on June 30, 2022 (NCT05445336); we updated the registry on December 18, 2023. Our study was reviewed and approved by the Hamilton Integrated Research Ethics Board (Project ID 14463).

### Study setting

Between January to June 2022, we sent emails to five organizations operating several LTC homes in Ontario; only two organizations were interested in participating in our study. One organization recruited one LTC home, while the second organization recruited two homes. We recruited one for-profit and two not-for profit homes, herein known as Home A, 120 beds, Home B, 425 beds, and Home C,240 beds. All three homes are located in southern Ontario, Canada in large metropolitan cities.

### Knowledge translation model

PREVENT is an organizational-level, service provision approach that embeds falls and fracture assessment and management strategies in LTC. The model utilizes several intervention functions and behaviour change techniques based on the Behaviour Change Wheel (Table 1) [20,22]. The PREVENT components can be found in S1 File.

PREVENT was designed to be integrated into regularly scheduled healthcare sessions (e.g., Professional Advisory Committee, Falls Committee), which may occur quarterly in LTC. We identified a local champion to help guide the implementation of the model and continue to promote best practices [21]. The standardized model

**Table 1. Intervention functions and behaviour change techniques used in PREVENT.**

| Intervention function | Definition of intervention function | Behaviour change technique | Behaviour change technique example |
|---|---|---|---|
| Education | Improving knowledge and understanding of roles and responsibilities for key healthcare professionals responsible for service provision and improving knowledge through a community of practice (21) | • Feedback on behaviour<br>• Prompts/cues<br>• Self-monitoring of behaviour | • Audit and Feedback Report |
| Training | Providing key professionals with core skills in management and leadership (21) | • Demonstration of the behaviour<br>• Instruction on how to perform a behaviour | • Train the trainer<br>• Educational videos |
| Modelling | Discussion of identified successful strategies for change within network (21) | • Demonstration of the behaviour | • Train the trainer<br>• Educational videos<br>• Training manual<br>• Fracture and Fall Prevention Guide |

includes: 1) train-the-trainer sessions with the local champion [23]; 2) a multidisciplinary educational session with fracture prevention care recommendations and real-world case studies [24]; 3) orientation to the Fracture Prevention Toolkit developed by the lead researcher (IBR) that is provided to each home to access the Fracture Risk Scale (FRS) which identifies residents at high risk of fracture (score between 4 to 8) and low risk of fracture (score between 1 to 3) [25], templates, standard order sets, and quick reference guide to the 2015 Canadian LTC fracture prevention guidelines; 4) an audit and feedback report developed by the local champion for physicians and healthcare providers regarding the number of residents at high versus low risk of fracture, number of residents who have experienced a fragility fracture in the last six months, and the number of high fracture risk residents on a fracture prevention medication (see S3 Table) [26]; and 5) implementation intentions that include an implementation intentions template and care planning template. We utilized a three-step process to implement PREVENT.

In step one (train-the-trainer), the research team (IBR, GI and LK) worked with each home to identify a local champion. We sought champions who the home considered trustworthy and influential, acted as a role model for behaviour change, supported and legitimized the work, provided a mechanism of communication between the research team and healthcare providers in LTC, and serviced to sustain long term gains [21]. The lead investigator (IBR) met with each local champion using an online platform (i.e., Zoom) and in-person at the LTC home to review how to deliver PREVENT, which included reviewing how to use the FRS to identify residents at high and low risk of fracture [25], a training manual, Fracture and Fall Prevention Guide, educational module, and other point of care tools. Our processes were guided by the Getting to Outcomes Framework, which includes a 10-step program for implementing, evaluating, and continuously improving prevention programs [27]. In step two, the local champion adapted the educational meeting to the LTC home and developed the audit and feedback report. During the third step, the local champion presented the adapted PREVENT material during the educational session to a leadership team consisting of physicians, nurse practitioners, registered nurses, dietitians, physiotherapists, kinesiologists, personal support workers, and interRAI coordinators. During the meeting, the leadership team also brainstormed and implemented intentions and care plans to identify and treat residents at high risk of fracture. LTC homes received remuneration for their participation in the study.

## Intervention timeline

| | Intervention Timeline |
|---|---|
| Month 1 and Month 2 | 1. LTC home recruitment<br>2. Identify local champion |
| Month 3 | 1. Train-the-trainer to prepare audit and feedback report and adapt/deliver the educational module |
| | **Interactive Session 1** |
| Month 4 | 1. Local champion leads education module with the leadership team<br>2. Audit and feedback report review<br>3. Develop action and care plan |
| Months 4 to Month 6 | 1. LTC home implements action and care plan |
| | **Interactive Session 2** |
| Month 7 | 1. Focus group with local champion and leadership team to understand barriers, facilitators, and adaptations<br>2. One-on-one interview with local champion to understand barriers, facilitators, and adaptations |

## Inclusion/Exclusion criteria

As this is a pragmatic study, our inclusion and exclusion criteria were broad and generalizable to represent real world practice. Our organizational level inclusion criterion was to include LTC homes with a minimum of 70 beds. At the individual level, we recruited one local champion in each home who was identified by the home's executive team and had most of the following characteristics: influence, ownership, physical presence at the point of change, grit, persuasiveness, and participative leadership style [25].

## Outcomes

**Primary outcome.** The primary outcome was feasibility of implementation defined by the number of LTC homes recruited within two months, length of time for the local champion to deliver the PREVENT program, and to determine the logistics of developing the audit and feedback report (Table 2) [28]. A fidelity checklist was developed by the lead investigator (IBR) and defined using a five-step guide proposed by Michie and colleagues [29,30]. The fidelity checklist assessed domains related to preparing the educational session (e.g., audit reporting, scheduling a date and time for the leadership team to meet), presenting the educational session (e.g., reviewing the FRS and guidelines, developing and reviewing the audit and feedback report), and leading discussions on the action and care plan processes.

Table 2. Feasibility outcomes and criteria for success [28].

| Feasibility outcomes | Criteria for success |
|---|---|
| Recruitment rate (organizational level–length of time to recruit three LTC homes) | • Recruit three LTC homes in Ontario within 2 months |
| Recruitment rate (participant level—length of time to identify and train local champion) | • Two months to identify, meet, and train local champion |
| Determine challenges to deliver the PREVENT model and program | • ≥ 80% on the fidelity checklist<br>• Identify facilitators of and barriers to implementation during the focus group |
| Determine challenges in linking the data for the audit and feedback report | • Identify facilitators of and barriers to developing the audit and feedback report |

The fidelity checklist included 23 items, and each item was rated from 0 (action was not completed), 0.25 (action was 25% completed), 0.50 (50% completed), 0.75, and 1.00. A total score of 85% (20/23) was considered good fidelity, moderate between 50% to 84%, and poor <50% [29,30]. The fidelity checklist was completed by the lead investigator (IBR) during each of the three education sessions. To understand adaptations after implementing the intervention, we held focus groups with the local champion and healthcare team. Focus groups were recorded and transcribed verbatim. Moreover, informal one-on-one interviews were held with each local champion using open-ended questions to identify adaptations to the model. These interviews were held three months after the educational session and were not recorded.

**Secondary outcomes.** Our secondary outcomes were to determine if there was a change in knowledge uptake and in the proportion of fracture prevention medication prescriptions before and after the intervention based on the audit report. We assessed change in knowledge uptake using a McNemar test. First, the research team developed a case study of a typical resident living in LTC. Next, healthcare providers were asked to complete a series of multiple-choice questions about identifying the residents FRS, prescribing medications, using calcium and vitamin D supplements, prescribing exercise, conducting an environmental hazard scan, and using hip protectors. Immediately following the education session, we asked healthcare providers to repeat the same multiple-choice questionnaire. Data on fracture prevention prescriptions were collected at baseline (six months prior to implementing the intervention) and post-intervention (three to four months after the intervention). We collected information on the number of residents at high risk or low risk of fracture using the FRS, the number of residents on a fracture prevention medication (e.g., bisphosphonates, denosumab), and the number of osteoporotic fractures at baseline and post-intervention. We defined fractures as any major osteoporotic fracture of the hip, pelvis, vertebrae, or distal radius [31,32]. A list of fracture prevention medications is provided in S3 Table.

## Statistical analysis

We reported our recruitment rates and fidelity scores using descriptive statistics as a total score or value. Results of the focus groups were reported using content analysis and the primary author (IBR) coded the data and met to discuss and compare the codes with two co-authors with experience in content analysis (LK or LH) [33]. Demographic characteristics of the local champion and leadership team were collected and reported using the number of individuals and a percentage. To assess change in knowledge uptake, we conducted a McNemar test for two related samples, for those who completed both pre- and post-assessments and reported the results using an exact significance (2-tailed). To report changes in fracture prevention medications, from the audit report, we conducted two data pulls at baseline and post-intervention; we included residents who "moved-in" (i.e., moved into the home during the post-intervention phase) in the analysis. We reported the results using a McNemar's Test. Quantitative results were analyzed using IBM SPSS Statistics for Windows, version 28 (Armonk, NY: IBM Corp) and qualitative transcripts were analyzed using NVivo, version 14 (QSR International Pty Ltd, Doncaster, VIC, Australia).

## Results

### Feasibility

We recruited three LTC homes between October 2022 and February 2023; our recruitment process was impacted by residual COVID-19 complications (Fig 1). After we identified a

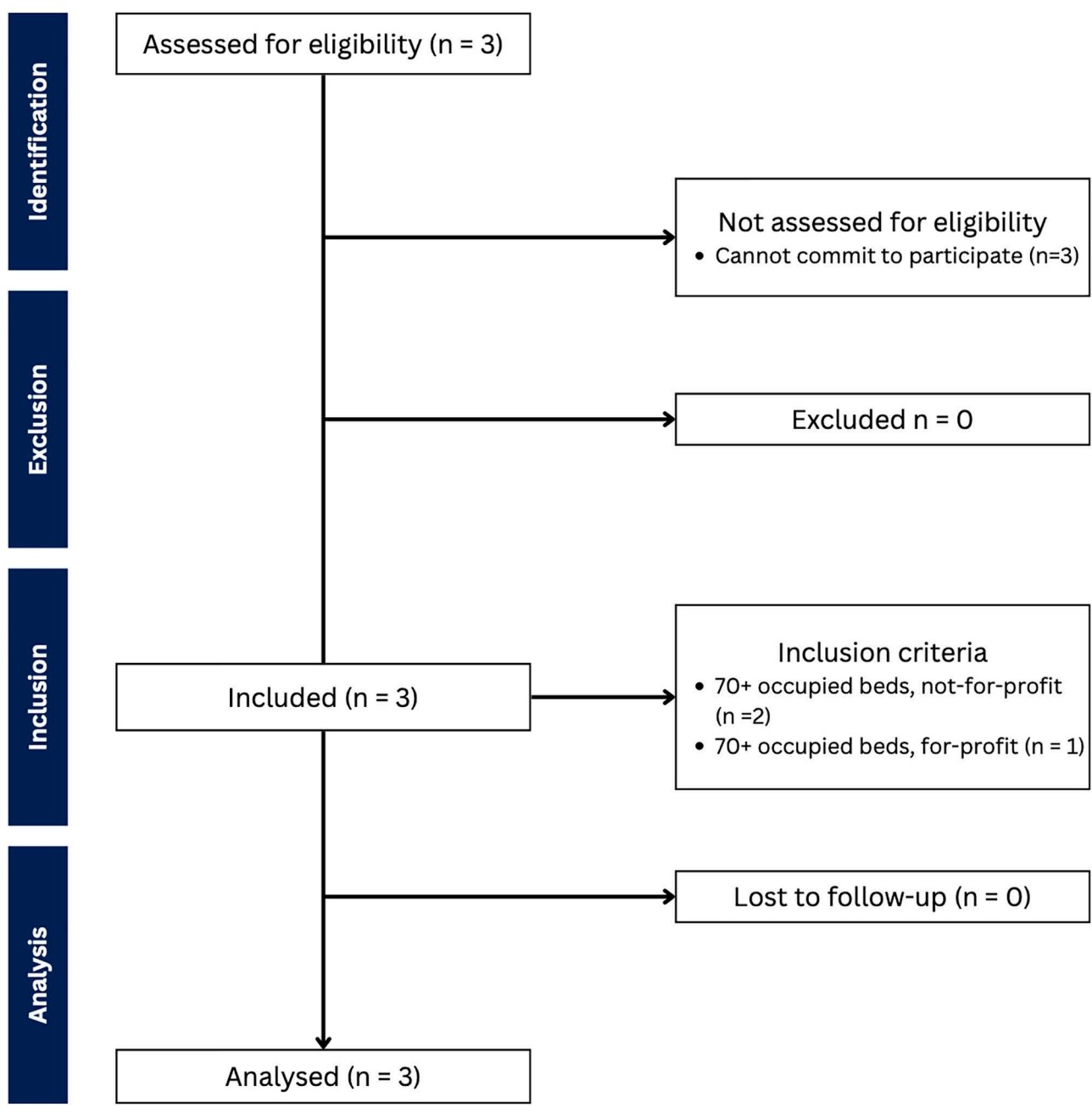

**Fig 1. STROBE flow chart.** STROBE, Strengthening the Reporting of Observational Studies in Epidemiology.

home, the home's administrative team took about a month to identify a potential local champion. Local champions included a kinesiologist and two nurse practitioners. Subsequently, our research team met with the local champion to complete the train-the-trainer sessions. Our research team completed two, one-hour train-the-trainer sessions with Home A; however, based on feedback from Home A, a third train-the-trainer session was added. The local champion led the third session where they practiced delivering the educational module to the research team. Homes B and C received three training sessions, each session was 1.5 hours

**Table 3. Local champion and leadership team characteristics (n = 3).**

| | Home A (k = 10) | Home B (k = 19) | Home C (k = 11) |
|---|---|---|---|
| **Local champion** | | | |
| Highest level of education | College | University/post-graduate | University/post-graduate |
| Fidelity score (total score 23 points) | 19.75/23 (86%) | 20.25/23 (88%) | 20.00/23 (87%) |
| **Leadership team** | | | |
| Sex (female), n (%) | 8 (80%) | 8 (42%) | 10 (91%) |
| Level of Education, n (%) | | | |
| College | 5 (50%) | 4 (21%) | 1 (9%) |
| University | 5 (50%) | 8 (42%) | 9 (82%) |
| Prefer not to answer | 0 (0%) | 7 (37%) | 1 (9%) |
| Role in LTC home, n(%) | | | |
| Physician | 0 (0%) | 4 (21%) | 1 (9%) |
| Pharmacist | 0 (0%) | 1 (5%) | 1 (9%) |
| Administrator | 1 (10%) | 1 (5%) | 0 (0%) |
| Director of care/Nursing director | 1 (10%) | 2 (11%) | 1 (9%) |
| Neighbourhood coordinator | 2 (20%) | 4 (21%) | 2 (18%) |
| Dietitian | 0 (0%) | 1 (5%) | 1 (9%) |
| Nurse practitioner | 0 (0%) | 1 (5%) | 2 (18%) |
| Nursing student | 1 (10%) | 0 (0%) | 0 (0%) |
| Registered nurse | 1 (10%) | 0 (0%) | 0 (0%) |
| Rehabilitation specialist/Physiotherapist | 2 (20%) | 2 (11%) | 2 (18%) |
| Personal support worker | 2 (20%) | 3 (16%) | 1 (9%) |
| Length of time worked within current profession in LTC, n (%) | | | |
| < 1 year | 2 (20%) | 1 (5%) | 1 (9%) |
| 1 to 2 years | 0 (0%) | 2 (11%) | 1 (9%) |
| 3 to 5 years | 2 (20%) | 1 (5%) | 1 (9%) |
| > 5 years | 5 (50%) | 8 (42%) | 7 (64%) |
| Prefer not to answer | 1 (10%) | 7 (16%) | 1 (9%) |
| Length of time providing care to residents at risk of falls and fractures, n (%) | | | |
| < 1 year | 2 (20%) | 1 (5%) | 1 (9%) |
| 1 to 3 years | 1 (10%) | 0 (0%) | 1 (9%) |
| 3 to 5 years | 6 (60%) | 10 (53%) | 8 (73%) |
| > 5 years | 1 (10%) | 1 (5%) | 0 (0%) |
| Prefer not to answer | 0 (0%) | 7 (16%) | 1 (9%) |

in length (see S4 Table). After the train-the-trainer sessions, the local champion required at least three months to review the educational materials, adapt the material to their home, and develop the audit and feedback report. The characteristics of the local champion and leadership team are presented in Table 3. PREVENT requires at least 10 to 12 months per home to recruit, train, and implement.

We had a diverse group of healthcare professionals on the leadership team (Table 3). In Home A, the local champion recruited ten healthcare professionals on the leadership team, in Home B, 19 healthcare professionals, and in Home C, 12 healthcare professionals. The attendance rate during the leadership team for Home A was 90% (9/10), Home B, 84% (16/19), and Home C, 90% (10/11). Fidelity scores ranged from moderate to good (Table 3).

## Challenges, successes, and adaptations

During the focus group, each home identified similar barriers to and facilitators of implementing PREVENT in practice (see S5 Table).

**Challenges.** Challenges to implementing PREVENT included having an unclear sustainability strategy, misperceptions between fall and fracture prevention strategies, fear of prescribing fracture prevention medications, and time barriers with completing the audit report. During the focus group, the local champion expressed that it was unclear who would sustain the role of implementing PREVENT after the research team had left; but Home C suggested the falls program leaders and/or staff educators should be involved upon study completion to sustain practices. The local champion cited lack of time and other commitments as a barrier to sustaining the model; all three champions noted that a major barrier was dedicating time to the audit and feedback report. To manually extract the data and link the variables to each resident for the audit report, the local champion required at least one full day per week for five weeks (i.e., 40 hours total). Moreover, several leadership members were unsure of the implementation plan after identifying a resident at high risk of fracture:

*"I found myself often sometimes ignoring the prompts for FRS <Fracture Risk Scale> unless there was a specific reason. Cold calling families to say your fracture risk scale is high, and that's the only reason I'm calling is weird."*

[Home B]

Several healthcare professionals on the leadership team felt the strategy to prevent fractures (i.e., PREVENT) was akin to their current fall prevention strategy. It was perceived that if a healthcare professional could prevent a fall, it would translate to preventing a fracture.

*"I think a lot of homes already are looking at all the PREVENT aspects, it's just not known as PREVENT or they're not grouping them together, but I think it's just something that's already being looked at from all the different disciplines."*

[Home A]

Other healthcare providers believed PREVENT was a tool to ensure the staff were performing their fall prevention duties, rather than a model of change to prevent fractures in LTC.

*"It's worked well just as a tool to let us know if there's any missing boxes that we're not checking for fracture prevention. It's very helpful with just making sure we're looking at everything and yea, just crossing off any things we can possibly to do help the residents."*

[Home A]

Additionally, several healthcare providers were unsure about the ethical implications of prescribing fracture prevention medications to residents at high risk of fracture. The main barrier to prescribing medications was fear of side effects (i.e., osteonecrosis of the jaw, atypical femur fractures, hypocalcemia). Some providers were also concerned about the ethical implications of prescribing fracture prevention medications to residents who could not consent or had limited knowledge of the medication.

"*We are worried that, we weren't overly confident that would lead to prescription. So there was a trigger. We didn't know if they were going to have informed consent and we didn't know if our nursing staff knew enough about these medications to provide it*"

[Home B].

**Successes.** Despite some challenges, we also identified several facilitators to our model. Facilitators included easy access to the FRS, clear and succinct educational material catered to each healthcare professional, and an educational module with an accredited program for physicians and nurses. All three homes deemed it beneficial to have the educational module delivered in person. We also received positive feedback on the type of information included in our educational module. In particular, healthcare providers found it helpful to review real-world case studies followed by discussions and suggestions to overcome barriers to implementation (e.g., using hip protectors for residents with incontinence).

"*A visual to see the walkers and those pieces, I think even the slide with the hip protectors, like how to apply them and which ones were incorrect. I think that's a nice reminder because I didn't even know how to position them correctly and I didn't even have awareness of the FRS. Like I kind of knew like our FRAT <Falls Risk Assessment Tool> score and our falls risk, and when I go to my units I know who the falls, the higher risk of falls are but I didn't, and then I just thought of them like okay, we have hip protectors for them, like what are we doing about it but I didn't necessarily do a full review of like let's look at their meds and let's look at their hips, are they wearing hip protectors, what are they, let's look at all these other things. I wasn't sort of looking at that.*"

[Home C].

**Adaptations.** The process of developing and implementing models into practice is an iterative process. Through this process, we recognized three adaptations to the PREVENT model including identifying additional characteristics for a successful local champion, developing an automated audit report, and creating tools to facilitate conversations for prescribing fracture prevention medications. It was suggested that the role of the local champion include co-championship with at least two healthcare professionals to help manage the implementation process. Suggested co-championship included one champion be either a registered nurse or a nurse practitioner and the second champion be a registered practical nurse or resident care manager. The benefit of including a resident care manager as a local champion is their established rapport with residents and caregivers, and their knowledge of daily practices and the types and frequency of assessments being done. Furthermore, the local champion must have a basic understanding of medications and have strong collaborations with other healthcare providers including physicians, nurses, dietitians, pharmacists, and physiotherapists. Additional co-champions may include a restorative care coach, RAI coordinator, and falls program lead. Home C described certain characteristics that may make a good local champion including someone who is interested and passionate about fracture prevention, willing to make change happen, and a good communicator. Other adaptations for future practice include working with the electronic medical record company to automate the audit report process, develop conversational toolkits (e.g., between nurses and residents) to aid dialogues about fracture prevention, and develop one-page educational guides for healthcare professionals and residents about the side effects of fracture prevention medications.

## Knowledge uptake

We found that the educational module improved knowledge uptake of the LTC recommendations among healthcare providers. Knowledge uptake increased by 25% (pre = 57/108 = 52.8% versus post = 84/108 = 77.8%, p-value < 0.01). A knowledge uptake subset analysis was performed to assess healthcare providers knowledge surrounding FRS scores, corresponding fracture risk, administrating fracture prevention medications and supplements (calcium and Vitamin D), using hip protectors, and appropriate exercise. Results of this analysis indicate that knowledge uptake also increased by 25% (pre = 50/99 = 50.5% versus post = 75/99 = 75.8%; p-value < 0.01).

Healthcare professional in Home B were prompted by the PREVENT model to develop and implement a "Fracture Prevention in LTC Order Set" as a multidisciplinary approach to care. Physicians, nurse practitioners, physiotherapists and dietitians used this order set at the time of study to review criteria for high fracture risk and non-pharmacological and pharmacological medical orders for residents living in their home. Furthermore, use of the order set will be rolled out to nurses who were identified as a key facilitator to integrate this tool into practice across the home. Changes in healthcare provider behavior and practice were also observed in Home C where providers reported greater awareness of the fracture risk scale and residents' fracture risk. They also shared that their pharmacist now reviews fracture risk during quarterly assessments and implemented a formal policy for hip protectors, which was previously lacking. Additionally, this home integrated the FRS Manual Calculation Tool into their admission process, ensuring that residents receive a fracture risk score upon admission.

## Audit report

The results of the audit report are shown in Table 4. Home A results indicated the number and percentage of residents at high risk of fracture on fracture prevention medications increased by 16.5% (pre = 19/97 = 19.6% versus post = 35/97 = 36.1%; p-value = 0.005). Non-statistically significant results were found for the number of residents at high risk of fracture changed by 6.2% (pre = 52/97 = 53.6% versus post = 58/97 = 59.8%; p-value = 0.41) and the number and percentage of residents who sustained a fragility fracture in the last six months did not change (pre = 3/97 = 3.1% versus post = 3/97 = 3.1%; p-value = 1.0). Home B reported an absolute change of 3% (31% at baseline and 34% post-intervention), while Home C increased by 11% (26% at baseline and 37% post-intervention).

## Discussion

Falls and hip fractures are a major health concern among older adults living in LTC [4,7–9]. The purpose of the iCARE study was to determine the feasibility (recruitment rate, barriers,

**Table 4. Aggregated pre-and-poster audit report.**

| | Homes A, B and C | |
| --- | --- | --- |
| | Pre-intervention | Post-intervention |
| Total number of residents in Home A, B, and C | 748 | 745 |
| # high fracture risk residents (%) | 430 (57%) | 433 (58%) |
| # low fracture risk residents (%) | 315 (42%) | 306 (41%) |
| # of residents that sustained a fragility fracture (%) | 22 (3%) | 28 (4%) |
| # high fracture risk residents on fracture medications (%) | 122 (28%) | 153 (35%) |

Note: The number of residents pre-and-post intervention are different because of the "moved-in" (i.e., moved into the LTC home) and "moved-out" (i.e., passed away).

facilitators, adaptations) of implementing PREVENT in three LTC homes in Ontario, Canada. The model includes a multifactorial intervention on diet and supplements, exercise, environmental adaptations, hip protectors, and medication reviews to treat residents at high risk of fracture [10]. The standardized model utilizes local champions, educational outreach methods, audit and feedback reports, and implementation intentions to change behaviour. Although we did not meet our recruitment criterion to recruit three homes in two months, our process was affected by residual COVID-19 complications. Nevertheless, it is possible to recruit LTC homes by accounting for a longer recruitment period (i.e., three homes over five months). We did meet our criterion to recruit a local champion and provide training within our timeframe. Our local champions successfully delivered PREVENT with a fidelity score ranging from moderate to good. We identified several barriers, facilitators, and adaptations to PREVENT. Benefits of the model include easy access to the FRS, clear and succinct educational material catered to each healthcare professional, and an accredited educational module for physicians and nurses. Few challenges such as misperceptions between the differences in fall and fracture prevention strategies, fear of prescribing fracture prevention medications, and time barriers with completing the audit report were identified. Our study also found an increase in knowledge regarding recommendations for preventing falls and fractures, which led to creation of new order sets and site-specific policies and implementation of the FRS manual calculation tool. An increase in the proportion of fracture prevention prescriptions post-intervention was also found. The development and implementation of knowledge translation models is an iterative process. This study identified additional changes to PREVENT including co-champions, automated audit reports, and tools to facilitate conversations when prescribing fracture prevention medications.

Effective and sustainable knowledge translation models are essential to encourage the uptake of evidence-based practices [12]. The greatest drawback is that there is limited evidence on effective and sustainable models in LTC practice and such models may not be generalizable from one setting to another [15]. Compared with community or acute care settings, there has also been limited implementation models for LTC, especially focused on fracture prevention. PREVENT is a scalable educational outreach model that utilizes a multifactorial intervention to reduce falls and fractures among residents in LTC [34]. Our study provides preliminary data that suggests PREVENT may be feasible within the LTC setting, particularly when structured to fit the unique practice environment and organizational structure. Most knowledge translation models have been evaluated in non-LTC settings [34]. A 2009 Cochrane review reported improvements in care that ranged from 4% to 12% using educational meetings, educational outreach programs, local champions, audit and feedback reports, and computerized reminders [34]. When designing PREVENT, we utilized such techniques (e.g., educational meetings, local champion, audit and feedback) as a method to change behaviour [21,24,26,34,35]. We observed improvements in the proportion of fracture prevention medications before and after the intervention, although some results were not statistically significant for specific homes. We saw greater improvements in prescriptions being filled in homes that had a lower rate of medication use. We specifically focused on fracture prevention prescriptions as the highest certainty of evidence to reduce the rate and risk of fractures is using such medications [17,18]. We hypothesize that an increased uptake of fracture prevention medications may translate to lower fracture rates in LTC; however, a large trial is needed to confirm the hypothesis.

Although we found a 7% increase in the number of prescriptions before and after the intervention; healthcare providers and residents expressed concerns with prescribing and taking fracture prevention medications, respectively. Physicians were concerned about polypharmacy and the potential effects of residents who are taking multiple medications, while

residents were more concerned about possible drug-related side effects than the consequences of fractures. Fear of osteoporosis drugs is a common concern among people living with osteoporosis. The National Osteoporosis Foundation conducted a survey with about 28,000 individuals of which 853 responded [36]. The foundation found that 38% of respondents had been prescribed a medication but refused to take it, with 79% stating that fear of side effects was the reason for refusing the medication [36]. Approximately 43% of survey respondents thought that the risk of side effects with osteoporosis treatment was greater than the benefit. Implementing models with behaviour change techniques will be important to target residents specifically. Potential adaptations to the PREVENT model may include targeting residents directly, in addition to healthcare workers, through an educational outreach program on the risks of fractures versus the benefits of taking an osteoporosis medication. The outreach program should be delivered by a potential resident council member that has developed rapport with residents in the LTC home.

Similar models to prevent fractures in LTC include the Bavarian Fall and Fracture Prevention model in Germany that focuses on exercise, documentation of falls, environmental adaptations, medication reviews, vitamin D, hip protectors, and education among healthcare providers [37,38]. The PREVENT model is a version of the Bavarian Fall and Fracture Prevention model adapted for the Ontario, Canada healthcare system. Akin to the Bavarian model, our model uses similar components including exercise, fall prevention, environmental adaptations, medication reviews, hip protectors, and educational material and all homes receive one-on-one training on how to implement the model. The Bavarian Fall and Fracture Prevention study is one of the few large studies to implement an educational outreach program to implement fracture prevention guidelines into LTC [37,38]. In the Bavarian study, the authors implemented a multifactorial educational outreach program and after one-year found an 18% reduction in hip fracture in residents from the intervention home (hazard rate ratio 0.82, 95% confidence interval 0.72–0.93) [37,38]. While the model was effective, it was a non-randomized, quasi-experimental design that utilized insurance company staff to facilitate and disseminate knowledge on fall and fracture prevention. The PREVENT model used components of the Bavarian model to adapt the model to the Canadian billing and healthcare system and reflect the differences between our healthcare systems. Our next steps will be to conduct a large randomized controlled trial (NCT04947722) of the adapted PREVENT model in 122 LTC homes across Ontario, Canada to determine if the model reduces the rate of hip fractures in residents at high risk of fracture. The ultimate goal will be to implement the model across the country. We hypothesize that an increased uptake of fracture prevention medications may translate to lower fracture rates in LTC; however, a large trial is needed to confirm our hypothesis.

Our PREVENT model attempted to leverage facilitators and limit barriers to implementation as described in other studies. A recent systematic review by the study team identified several barriers to implementing guidelines into LTC settings including time constraints and inadequate staffing, cost and lack of resources, knowledge gaps, and lack of teamwork and organizational support [39]. Our model addressed such barriers including time constraints, lack of resources, and knowledge gaps by developing a multi-modal approach to embed a validated computer decision support tool and the FRS into current established care pathways. PREVENT also provides an accredited Continuing Medical Education module to physicians and nurses as a method to disseminate evidence-based knowledge. We worked with the home's management team to identify potential local champions who could help lead the educational module and sustain best practices after the research team left. Characteristics for an effective local champion were identified based on previous literature [21,40]. Through the iCARE study, we identified additional characteristics to identify a local champion including

having co-champions manage the implementation process and developing conversational toolkits to aid dialogues between healthcare providers and residents on fracture prevention medications. We are also working with the Canadian electronic medical record company to develop an automated audit and feedback report to address the time barrier identified by homes. We suggest that future knowledge translation and implementation science researchers look to our adapted PREVENT model as there may be important changes to help move evidence into practice.

## Strengths and limitations

The iCARE study had several methodological strengths including the recruitment of homes that were geographically diverse, of varied profit status, and located in communities of varied population sizes. Our study implemented a multifactorial intervention that engaged a wide variety of healthcare providers. In addition, our study builds on previous knowledge and was designed to leverage facilitators and limit barriers to implementation. Given that our intervention was multifaceted, it was challenging to identify the most significant components (e.g., educational outreach, local champion, audit and feedback report) that would result in effective implementation of PREVENT. Moreover, measures of organizational context such as work culture and leadership style, were not captured in the iCARE study which could have a significant impact on our model's components. For this study, we did not involve residents or their caregivers during the implementation process; however, the Family Councils Ontario and Ontario Association of Residents' Councils advised upon the 2015 guidelines and the development of the PREVENT model. Future adaptations for the PREVENT model will be to include end-users on the research team and have LTC residents join a Resident Advisory Committee to review and provide feedback on study-related materials to aid the implementation process. Lastly, we did not collect secondary outcomes including changes in diet, exercise, environmental adaptations, and hip protector adoption due to the limited resources in the iCARE study. For the larger PREVENT trial, we will collect these secondary outcomes to gain a larger perspective of the changes the PREVENT model may provide.

## Conclusion

PREVENT is a multifactorial educational outreach model to implement education on diet, exercise, environmental adaptations, hip protectors, medications (including calcium and vitamin D) and medication reviews to treat residents at high risk of fracture. The model utilizes educational meetings, local champions, and audit and feedback as methods to promote behaviour change. The model requires ten months to recruit, train, and implement PREVENT, with five of the ten months dedicated toward recruitment of LTC homes. We suggest co-championship when delivering multifactorial models in LTC and developing tools that can be easily embedded into routine practice. Our study found an increase in knowledge uptake and the proportion of fracture prevention prescriptions post-intervention. Our next steps will be to lead a large, randomized controlled trial to implement PREVENT to determine if the model reduces the risk and rate of hip fractures in residents living in LTC.

## Supporting information

**S1 File. PREVENT components and modifications.**
(PDF)

**S3 File. Deidentified data for replication.**
(XLSX)

**S1 Table.  STROBE checklist.**
(PDF)

**S2 Table.  TIDieR checklist.**
(PDF)

**S3 Table.  Fracture prevention medications.**
(XLSX)

**S4 Table.  Feasibility outcome results.**
(PDF)

**S5 Table.  Full list of themes.**
(PDF)

## Author contributions

**Conceptualization:** Isabel B. Rodrigues, George Ioannidis, Caitlin McArthur, Alexandra Papaioannou.

**Data curation:** Isabel B. Rodrigues, Lauren L. Kane.

**Formal analysis:** Isabel B. Rodrigues, George Ioannidis, Lauren L. Kane.

**Funding acquisition:** Isabel B. Rodrigues, George Ioannidis, Loretta M. Hillier, Jonathan Adachi, George Heckman, John Hirdes, Jayna Holroyd-Leduc, Susan Jaglal, Sharon Kaasalainen, Sharon Marr, Caitlin McArthur, Sharon Straus, Jean-Eric Tarride, Andrew P. Costa, Lehana Thabane, Alexandra Papaioannou.

**Investigation:** Isabel B. Rodrigues, George Ioannidis, Loretta M. Hillier.

**Methodology:** Isabel B. Rodrigues, George Ioannidis, Loretta M. Hillier, Andrew P. Costa, Alexandra Papaioannou.

**Project administration:** Isabel B. Rodrigues, George Ioannidis, Lauren L. Kane, Momina Abbas, Alexandra Papaioannou.

**Resources:** Isabel B. Rodrigues, George Ioannidis, Caitlin McArthur, Momina Abbas, Alexandra Papaioannou.

**Software:** Isabel B. Rodrigues.

**Supervision:** George Ioannidis, Alexandra Papaioannou.

**Validation:** Isabel B. Rodrigues, George Ioannidis, Lauren L. Kane.

**Visualization:** Isabel B. Rodrigues, George Ioannidis, Lauren L. Kane.

**Writing – original draft:** Isabel B. Rodrigues.

**Writing – review & editing:** Isabel B. Rodrigues, George Ioannidis, Lauren L. Kane, Loretta M. Hillier, Jonathan Adachi, George Heckman, John Hirdes, Jayna Holroyd-Leduc, Susan Jaglal, Sharon Kaasalainen, Sharon Marr, Caitlin McArthur, Sharon Straus, Jean-Eric Tarride, Momina Abbas, Andrew P. Costa, Arthur N. Lau, Lehana Thabane, Alexandra Papaioannou.

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
