## [Decision Letter · Decision Letter 0]

29 May 2024

PGPH-D-24-00466

The iCARE feasibility non-experimental design study: An integrated collection of education modules for fall and fracture prevention for healthcare providers in long term care

Dear Dr. Rodrigues,

Thank you for submitting your manuscript to PLOS Global Public Health. After careful consideration, we feel that it has merit but does not fully meet PLOS Global Public Health’s publication criteria as it currently stands. Therefore, we invite you to submit a revised version of the manuscript that addresses the points raised during the review process.

Please respond to these important comments and revise your manuscript accordingly. Importantly, make sure to highlight all changes to the manuscript draft in YELLOW to allow for further assessment of the same.

We look forward to receiving your revised manuscript.

Kind regards,

Professor Razak M Gyasi, PhD, PD

Academic Editor

Journal Requirements:

1. Please provide separate figure files in .eps or .eps format only and remove any figures embedded in your manuscript file. Please also ensure all files are under our size limit of 10MB.

2. In the online submission form, you indicated that "Data available upon request from corresponding author".

3. Uploaded as supplementary information.

Additional Editor Comments (if provided):

Reviewers' comments:

Reviewer's Responses to Questions

**Comments to the Author**

1. Does this manuscript meet PLOS Global Public Health’s publication criteria ? Is the manuscript technically sound, and do the data support the conclusions? The manuscript must describe methodologically and ethically rigorous research with conclusions that are appropriately drawn based on the data presented.

Reviewer #1: Yes

Reviewer #2: Yes

Reviewer #3: Yes

Reviewer #4: Yes

Reviewer #5: Yes

Reviewer #6: Yes

Reviewer #7: Yes

2. Has the statistical analysis been performed appropriately and rigorously?

Reviewer #1: N/A

Reviewer #2: Yes

Reviewer #3: Yes

Reviewer #4: I don't know

Reviewer #5: Yes

Reviewer #6: N/A

Reviewer #7: Yes

3. Have the authors made all data underlying the findings in their manuscript fully available (please refer to the Data Availability Statement at the start of the manuscript PDF file)?

Reviewer #1: Yes

Reviewer #2: Yes

Reviewer #3: Yes

Reviewer #4: Yes

Reviewer #5: Yes

Reviewer #6: No

Reviewer #7: Yes

4. Is the manuscript presented in an intelligible fashion and written in standard English?

Reviewer #1: Yes

Reviewer #2: Yes

Reviewer #3: Yes

Reviewer #4: No

Reviewer #5: Yes

Reviewer #6: Yes

Reviewer #7: Yes

5. Review Comments to the Author

Reviewer #1: I enjoyed reading the article, and I commend the attention to detail. The authors aimed to determine the feasibility of implementing the PREVENT model in long-term care homes, focusing on falls and hip fractures. Proposed or reported benefits of the model include easy access to the Fracture Risk Scale, clear educational material, and an accredited educational module. Challenges included misperceptions, fear of side effects, and time barriers. The study did not increase knowledge uptake of the guidelines.

Here are my comments:

In the study setting, you noted that 2 out of 5 LTc agreed, yet you worked with 3. How did you achieve this?

Comparing the preaudit and post-audit reports, changes appear to be minimal; a p value could have helped identify if the changes were at least statistically significant.

Setting the stage for a much larger project, the author might wish to add a recommendation section on what can be done to improve the efficacy of PREVENT in reducing falls and hip fractures rather than mentioning it in passing.

The authors noted some existing models, such as the Bavarian Fall and Fracture Prevention model in Germany, and how such a model might not fit the Canadian system. It's important for the authors to also note whether the PREVENT model can be used outside of the Canadian health system and even if such a model is useful in all Canadian provinces.

Reviewer #2: The research entitled “The iCARE feasibility non-experimental design study: An integrated collection of education modules for fall and fracture prevention for healthcare providers in long-term care” was conducted among healthcare providers of three long-term care (LTC) facilities in Ontario, Canada. The iCARE study was conducted to see the feasibility of implementing the PREVENT model to educate health workers working in LTC facilities on how to prevent falls and injuries in residents of LTC facilities. The manuscript is well-written and easy to follow. However, my great discomfort lies in the use of iCARE in the title of the paper. I have provided some input so that the paper can be made better.

1. The title says the study is about iCARE intervention. However, it has been explained very nominally in the paper. Instead, another intervention (PREVENT) has been described in much detail; however, iCARE should be explained in more detail. It is not clear how iCARE differs from PREVENT.

When reading the whole manuscript, the intervention was PREVENT, and iCARE was the project that administered PREVENT. Hence, I suggest that the authors make a distinction between these two things. If iCARE is a project, I believe explaining that in a study setting and removing it from the title would make more sense.

2. There is no result of the t-test in the result section. I recommend that the authors include bivariate results of the t-test, including t-statistics, p-value, and mean difference.

3. On page 13, line no. 312, Are you talking about “mean difference” instead of “mean?”

4. On page 14, line no. 316, authors have reported odds ratio. However, no regression techniques have been mentioned in the methods section. So, I am wondering how the authors got the odds ratio. Hence, I suggest that the authors review their quantitative data analysis techniques.

5. In the result section, It would be nice to see a joint display of some common variables in quantitative and qualitative data. For example, you can display a reduction in osteoporosis medication from QUANT and justify that from QUAL data.

6. In the discussion section, I could not see the link between the Fidelity score and the knowledge score of long-term care medical staff. Is it possible to think in this direction as well?

Thank you.

Reviewer #3: The study evaluates the feasibility of implementing the PREVENT model, a multifactorial intervention to prevent fractures in long-term care (LTC) residents. Utilizing a mixed method strengthens the study. While the increase in osteoporosis medication prescriptions is a positive finding, the lack of statistically significant knowledge uptake seeks further investigation.

Reviewer #4: Thank you for the opportunity to review this manuscript. the study is really a good initiative with the current increase in elderly/dependancey. Anyway I have listed some comments to increase the quality of the manuscript.

Please avoid too much text within brackets.

better avoid the "We", "our"

Give emphasis on maintaining the flow of manuscript:Describe the results according to the stated objectives, then discussion for your results and finally conclusion. Please include some justifications for "no statistically significant increase" rather than simply stating proportions.

The discussion part of the manuscript need to be redrafted with some comparisons of previous similar studies

Reviewer #5: This study is apt! It is aligning with a contemporary prevention algorithm in the area of reducing falls especially among elderly. This study has public health significance and implications. The tools used for the purpose of training and data collections are in tandem with the title and study protocols. The findings are also evidenced-based. The Model adopted actually fit the study. The outcome of the study adds to the existing body of knowledge and will also serve as a further benchmark for other studies.

Reviewer #6: The paper was interesting and well described regarding the feasibility study of implementing an intervention. However, it would be good to additionally explain the relation of iCARE and PREVENT.

In result section, it would be bettter understandable if the recruitment outcomes (time to recruit LTC homes, time consumed for recruiting local champion, and time spent for deliver knowledge to care team, time used for providing audit and feedback repot) were explicitly demonstrated in table. Eventually, the explanation of getting to outcomes according to the using framework should be done. Moreover, the machanism of change in health care provider behavior according to PREVENT model should also be elaborated.

Regarding to secondary outcomes, please explain why there was no chnage in knowledge uptake but have change on prescription of medication. Should there be any other secondary outcomes regarding diet, excercise, environmental adaptation, and hip protector adoption.

Please provide further discussion on the relation of fear on side effects of medication and increasing of medication prescription.

Reviewer #7: Firstly, I would like to highlight the great value of the study's aim and results. The technical standard of the approach appears to be excellent; it thoroughly describes the details of the iCARE study, PREVENT, and their barriers and facilitators.

The publication holds significant value, and to continue fostering transparency, collaboration, and innovation, I strongly recommend making the dataset and codes available on a repository. This action does not prevent authors from sharing data upon request. However, sharing not only enhances the credibility and reproducibility of research but also accelerates scientific progress by enabling interdisciplinary collaboration and new discoveries.

By the end of the discussion, authors do recognize that residents or family caregivers were not considered during the implementation process. I believe this limitation extends beyond the implementation process. Residents' experiences and opinions— their voices—should be considered from the planning of the approach and study through all steps. Particularly in LTC, where it is common to neglect the autonomy of people who live in these facilities, a program developed to prevent an event as common as a fall among them needs to consider the impacted people. Their experiences should also inform the design and serve as a source of evaluation about the approach.

For instance, in the third step of PREVENT, when the local champion presents the adapted material during the leadership team meeting with various professionals, some residents could have been included. I understand that the involvement of LTC residents in advocating for their needs depends on the management model adopted by the facility. However, as research takes place, it is a responsibility to promote opportunities for an integrated approach to legitimize people's autonomy, as PREVENT's goal is to prevent falls and fractures among LTC residents. It's only understandable that they are also involved.

One of the goals of the iCare study centered on training trainers, specifically the local champion, which is a great strategy. I recommend sharing more details on the motivation, challenges, and strategies for selecting the local champion and their experience on this role. Understanding this role's experience seems to be a key component for the sustainability of the approach in LTC.

The manuscript reports absolute changes in outcomes, which raises questions about statistical rigor, particularly concerning sample sizes of certain groups. I recommend using appropriate statistical tests to compare differences between groups, especially when dealing with smaller sample sizes.

I agree with the authors that effective and sustainable knowledge translation models are essential to encourage the uptake of evidence-based practices, and that it takes time. However, I would like to share some concerns if a randomized controlled trial to implement PREVENT will take place in 122 homes. The limitations presented in this paper will need to be addressed, such as: i) using an automated system to monitor and evaluate the educational impact; ii) addressing the potential flaw in sustaining the program's success on medication prescription when polypharmacy is a big concern in LTC. Perhaps a multivariate analysis could be conducted to include other aspects of a person's eligibility to have osteoporosis medication prescribed; and as mentioned, iii) involving residents from the planning of the approach and study through all steps, including monitoring and evaluation.

6. PLOS authors have the option to publish the peer review history of their article (what does this mean? ). If published, this will include your full peer review and any attached files.

**Do you want your identity to be public for this peer review?** For information about this choice, including consent withdrawal, please see our Privacy Policy .

Reviewer #1: **Yes: ** Gbolahan Olatunji

Reviewer #2: No

Reviewer #3: **Yes: ** Ripiye Nanna Rebecca

Reviewer #4: No

Reviewer #5: **Yes: ** Khadijat Toyin Musah

Reviewer #6: No

Reviewer #7: No

---

## [Decision Letter · Decision Letter 1]

23 Sep 2024

Assessing the feasibility of an integrated collection of education modules for fall and fracture prevention (iCARE) for healthcare providers in long term care: A longitudinal study

PGPH-D-24-00466R1

Dear Dr Rodrigues,

We are pleased to inform you that your manuscript 'Assessing the feasibility of an integrated collection of education modules for fall and fracture prevention (iCARE) for healthcare providers in long term care: A longitudinal study' has been provisionally accepted for publication in PLOS Global Public Health.

If your institution or institutions have a press office, please notify them about your upcoming paper to help maximize its impact. If they'll be preparing press materials, please inform our press team as soon as possible -- no later than 48 hours after receiving the formal acceptance. Your manuscript will remain under strict press embargo until 2 pm Eastern Time on the date of publication. For more information, please contact globalpubhealth@plos.org .

Best regards,

Professor Razak M Gyasi, PhD, PD

Academic Editor

Editors comment:

Please remove the acronym in the title and define it in full in the abstract instead.

Reviewer Comments (if any, and for reference):

Reviewer's Responses to Questions

**Comments to the Author**

1. If the authors have adequately addressed your comments raised in a previous round of review and you feel that this manuscript is now acceptable for publication, you may indicate that here to bypass the “Comments to the Author” section, enter your conflict of interest statement in the “Confidential to Editor” section, and submit your "Accept" recommendation.

Reviewer #1: All comments have been addressed

Reviewer #2: All comments have been addressed

2. Does this manuscript meet PLOS Global Public Health’s publication criteria ? Is the manuscript technically sound, and do the data support the conclusions? The manuscript must describe methodologically and ethically rigorous research with conclusions that are appropriately drawn based on the data presented.

Reviewer #1: Yes

Reviewer #2: Yes

3. Has the statistical analysis been performed appropriately and rigorously?

Reviewer #1: Yes

Reviewer #2: Yes

4. Have the authors made all data underlying the findings in their manuscript fully available (please refer to the Data Availability Statement at the start of the manuscript PDF file)?

Reviewer #1: Yes

Reviewer #2: Yes

5. Is the manuscript presented in an intelligible fashion and written in standard English?

Reviewer #1: Yes

Reviewer #2: Yes

6. Review Comments to the Author

Reviewer #1: Comments addressed.

Reviewer #2: Thank you very much for addressing my all comments. Now, the paper looks nice. Best wishes for the publication and future research.

7. PLOS authors have the option to publish the peer review history of their article (what does this mean? ). If published, this will include your full peer review and any attached files.

**Do you want your identity to be public for this peer review?** For information about this choice, including consent withdrawal, please see our Privacy Policy .

Reviewer #1: **Yes: ** Gbolahan Olatunji

Reviewer #2: No
